

# Is it worth the extra mile? Comparing environmental DNA and RNA metabarcoding for vertebrate and invertebrate biodiversity surveys in a lowland stream

Till-Hendrik Macher[1,2], Jens Arle[3], Arne J. Beermann[1,4], Lina Frank[1], Kamil Hupało[1], Jan Koschorreck[3], Robin Schütz[1] and Florian Leese[1,4]

[1] Aquatic Ecosystem Research, University of Duisburg-Essen, Essen, Germany
[2] Biogeography, University of Trier, Trier, Germany
[3] German Environment Agency, Berlin, Germany
[4] Centre for Water and Environmental Research (ZWU), University of Duisburg-Essen, Essen, Germany

## ABSTRACT

Environmental DNA (eDNA) metabarcoding has emerged as a promising approach to assess biodiversity and derive ecological status classes from water samples. However, a limitation of eDNA surveys is that detected DNA molecules may originate from other places or even dead organisms, distorting local biodiversity assessments. Environmental RNA (eRNA) metabarcoding has recently been proposed as a complementary tool for more localized assessments of the biological community. In this study, we evaluated the effectiveness of eDNA and eRNA metabarcoding for inferring the richness and species distribution patterns of vertebrates and invertebrates in a Central European lowland river. We collected water samples and analyzed them using a 12S marker for vertebrates and a COI marker for invertebrates. We detected 31 fish, 16 mammal, 10 bird and one lamprey species in the vertebrate dataset. While results were largely consistent, we detected a higher number of species when analysing eRNA (mean = 30.89) than eDNA (mean = 26.16). Also, eRNA detections had a stronger local signature than eDNA detections when compared against species distribution patterns from traditional fish monitoring data. For invertebrates, we detected 109 arthropod, 22 annelid, 12 rotiferan, eight molluscan and four cnidarian species. In contrast to the pattern of vertebrate richness, we detected a higher richness using eDNA (mean = 41.37) compared to eRNA (mean = 22.42). Our findings primarily show that eDNA and eRNA-based detections are comparable for vertebrate and invertebrate taxa. Biological replication was important for both template molecules studied. Signal detections for vertebrates were more localized for eRNA compared to eDNA. Overall, the advantages of the extra steps needed for eRNA analyses depend on the study question but both methods provide important data for biodiversity monitoring and research.

Corresponding authors
Till-Hendrik Macher,
macher@uni-trier.de
Florian Leese,
florian.leese@uni-due.de

## INTRODUCTION

Biodiversity surveys are the basis for ecological research as well as biological monitoring programmes to quantify ecosystem integrity and derive trends in a rapidly changing world. Environmental DNA (eDNA) metabarcoding has recently emerged as a promising complementary approach to traditional surveys for assessing diversity and ecological status (*Hering et al., 2018*; *Pont et al., 2021*). Here, DNA traces can be extracted from environmental samples such as water (*Hänfling et al., 2016*; *Pont et al., 2022*), soil (*Fahner et al., 2016*; *Rota et al., 2020*), air (*Roger et al., 2022*; *Lynggaard et al., 2022*) or rainwater (*Macher et al., 2023a*) without having to first isolate or capture individual organisms. Although eDNA metabarcoding offers an efficient and non-invasive method to detect vertebrate (*Sales et al., 2020*; *Macher et al., 2021*), invertebrate (*Mächler et al., 2016*; *Blackman et al., 2022*) and diatom (*Kulaš et al., 2022*) species, methodological challenges remain. In particular, uncertainties about the state and origin of environmental DNA are a concern (*Cristescu & Hebert, 2018*). Environmental DNA may persist in water from days to more than a month (*Dejean et al., 2011*; *Mauvisseau et al., 2021*), can be transported over distances of up to 10 km (*Deiner & Altermatt, 2014*; *Wacker et al., 2019*) and can be retained in and resuspended from the sediment (*Shogren et al., 2017*). Furthermore, eDNA signals can also derive from decaying specimens (*e.g.*, fish carcasses), bird faeces (*e.g.*, migratory birds) or wastewater effluents (*e.g.*, commercial fish), leading to false positive detections of species.

Recently, environmental RNA (eRNA) metabarcoding has been proposed as a highly promising new approach to accurately recover the local living community (*Cristescu, 2019*; *Yates, Derry & Cristescu, 2021*). Environmental RNA refers to RNA molecules that are released by living organisms into their environment. Like eDNA, the term eRNA is specifically used for RNA that is present in the environment in the form of cells, vesicles, or free RNA. RNA is a substantially less stable molecule than DNA (*Fontaine & Guillot, 2003*) and therefore degrades faster. For example, eRNA has a 4–5 h shorter half-life compared to eDNA and is released in higher copy numbers than eDNA (*Marshall, Vanderploeg & Chaganti, 2021*). These properties may make eRNA advantageous over eDNA for analyses of local community compositions (*Cristescu, 2019*). To capitalise on these potential strengths for bioassessment, the limited existing knowledge on eRNA's behaviour in the environment needs to be further explored. Previous studies have already investigated eRNA-based identification to investigate fish (*Miyata et al., 2021*), freshwater algae, arthropods (*Miyata et al., 2022*), unicellular eukaryotes, bacterial communities, foraminiferans and macrofauna (*Pochon et al., 2015*; *Pochon et al., 2017*; *Laroche et al., 2018*; *Greco et al., 2022*; *Giroux et al., 2022*). However, the potential of eRNA metabarcoding as a method for routine monitoring has not yet been sufficiently explored.

Our study addresses two critical aspects to improve our understanding of genetic signals derived from aquatic ecosystems: the temporal distribution of eDNA and eRNA signals and the required number of samples for accurate biodiversity estimates. Although existing studies have highlighted the inhomogeneous distribution of eDNA signals and the influence of sampling efforts on species richness (*Cantera et al., 2019*; *Beentjes et al., 2019*; *Macher*

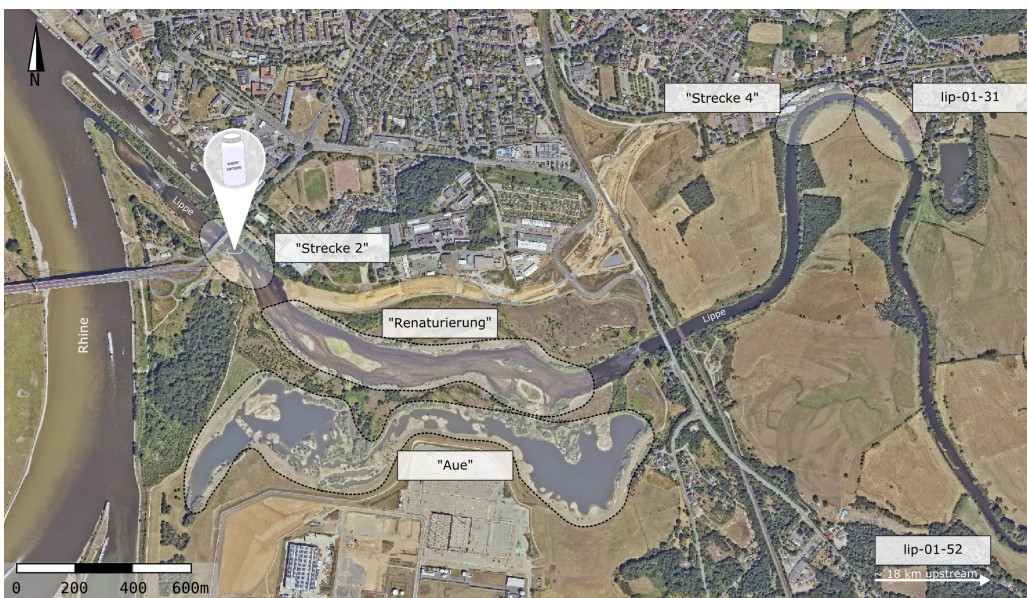

**Figure 1** **Sampling sites at the river Lippe near Wesel, Germany.** In total, 19 water samples were collected at a single site in the river mouth. Six electrofishing sites for restoration monitoring (EGLV) and WFD monitoring (LANUV) that have been assessed in 2016 and 2019 are outlined. The sites "Strecke 2", "Renaturierung", and "Aue" were classified as downstream sites. The sites "Strecke 4", lip-01-31, and lip-0-52 were classified as upstream sites. Source: Orthophotos Geobasis NRW, TIM-online 2.

*et al., 2021*), comparative data on these aspects for eDNA and eRNA is lacking. Thus, we conducted an eDNA and eRNA metabarcoding comparison spanning 19 water samples (3 L per sample) collected at 5-minute intervals from the same site. By evaluating both vertebrate and invertebrate communities, our study explicitly compared the efficiency of both approaches in biodiversity monitoring applications of these important bioindicators. Specifically, our aim was to compare species richness, assess species detection rates, evaluate signal locality and community change across a 1.5 h time interval for eDNA and eRNA. Based on the data, we offer recommendations for optimal sampling efforts in biodiversity monitoring, particularly by scrutinizing the utility of eRNA samples.

## MATERIALS AND METHODS

### Sampling

The sampling site was located at the river Lippe mouth, near Wesel, Germany (51°38′47.3″N 6°36′43.6″E; Fig. 1; Orthophotos Geobasis NRW, TIM-online 2). Here, the river Lippe is classified as river type 15 (sand and clay characterised lowland rivers). The river Lippe mouth was restored in the year 2014 and now possesses diverse habitat structures, with fast- and slow-flowing sections, riverbanks, meadows, and floodplains (see Fig. 1, "Strecke 2", "Renaturierung", and "Aue"). The upstream sections exhibit a less diverse habitat structure, characterized by artificial straightening, lacking floodplains, and devoid of oxbow lakes. ("Strecke 4" and site lip-01-31).

Samples were obtained on 13th July 2021. All sampling equipment was sterilized in the lab using a self-made decontamination solution (DIY-DS, 0.6% bleach, 1% NaOH, 1% Alconox, 90 mM sodium bicarbonate; *Buchner, Haase & Leese, 2021*). In total, 19 water samples were collected at a single site using sterile 5 L buckets. Samples were collected as close to the middle of the stream as possible to minimize spatial effects of riverbanks and ensure a steady flow velocity. Samples were collected at 5 min intervals, leading to a time series spanning 1.5 h. All water samples were filtered on-site using Smith-Root eDNA self-preserving filter packs (1.2 µm; Smith-Root, Vancouver, WA, USA) attached to a Vampire sampler peristaltic pump (Bürkle, Bad Bellingen, Germany). In total, 3 L of water was filtered per sample (total filtered volume of 57 L). One filter was used to filter 1 L of deionized water as a field negative control (= field blank). The filter membrane was removed from the filter capsule using sterile forceps immediately after water filtration. The filter membranes were transferred into new 1.5 mL Eppendorf tubes and stored on dry ice. After returning to the lab, all samples were stored in a −80 °C freezer overnight.

## Lysis and total nucleic acid extraction

All wet-lab steps were conducted under sterile conditions in a dedicated sterile laboratory (UV lights, sterile benches, overalls, gloves, and face masks). For each filter, a twist-top tube with one spoon of about 10 small (1 mm) and 20 large (2 mm) zirconia beads was prepared. Subsequently, filters were taken from the −80 °C freezer and chopped into small pieces in sterile Petri dishes using sterile forceps. The filter pieces were transferred into the twist-top tubes. Then, 1 mL of 4 M GITC lysis buffer was added, and samples were bead-beaten for 5 min at 2,400 rpm (Mini BeadBeater 96; Biospec Products, Bartlesville, OK, USA). Afterwards, samples were centrifuged for 3 min (11,000 × $g$).

A premix containing 20 µL of magnetic beads (SeraSil-Mag 400 silica-coated beads), 120 µL of TE minimum (pH 8; *Buchner, 2022a*), and 280 µL of isopropanol for each sample was prepared (420 µL per sample). The premix was distributed into 1.5 mL Eppendorf tubes and 210 µL of the cleared lysate was added. All samples were mixed by pipetting up and down 10 times. Subsequently, samples were incubated at 20 °C for 5 min at 1,000 rpm. Beads were attracted to a custom-built magnet for 2 min and the supernatant was discarded. Two wash steps were performed by adding 150 µL of wash buffer (10 mM Tris, 80% EtOH, pH 7.5; *Buchner, 2022a*) and mixing by pipetting 10 times, followed by incubation (5 min at 20 °C and 1,000 rpm), magnetically attracting the beads, and discarding the supernatant. After the second washing step, the beads were dried for 5 min at 50 °C to remove traces of ethanol. Lastly, samples were supplemented with 55 µL of elution buffer (10 mM Tris, pH 8.5; *Buchner, 2022a*), mixed by pipetting 10 times, and incubated (2 min at 1,000 rpm and 20 °C). After the final incubation, beads were attracted to the magnet for 2 min and 50 µL of the supernatant containing the purified total nucleic acid (TNA) was transferred to a new 1.5 mL Eppendorf tube. At this point, the TNA was split into a DNA fraction and an RNA fraction. The DNA fraction was directly used for polymerase chain reaction (PCR), while the RNA fraction first required DNAse treatment and reverse transcription.

## DNAse treatment and reverse transcription

Initially, each 8 µL sample of the RNA fraction was mixed with 1 µL of ezDNA buffer and 1 µL of ezDNase enzyme (ThermoFisher Scientific, Dreieich, Germany) in PCR strip tubes placed on a cooling rack. Samples were vortexed briefly and then incubated in a thermocycler (Biometra TAdvanced 96; Analytik Jena, Jena, Germany) for 2 min at 37 °C (with subsequent cooling to 4 °C until the samples were taken out). Afterwards, 4 µL of Superscript IV VILO Master Mix (ThermoFisher Scientific) and 6 µL of PCR-grade water were added to each sample (on a cooling rack). Samples were vortexed briefly and incubated in the thermocycler for 10 min at 25 °C, 10 min at 50 °C and 5 min at 85 °C (with a subsequent pause at 16 °C until samples were taken out) to transcribe the RNA into cDNA. Both the DNA fraction and RNA fraction were visualized on a 1% agarose gel. All samples were stored at −20 °C overnight until PCR amplification.

## DNA and RNA amplification and sequencing

A two-step PCR approach was applied to amplify the extracted DNA and transcribed RNA to target invertebrates using the fwh primer pair fwhF2 and fwhR2n (*Vamos, Elbrecht & Leese, 2017*) and vertebrates using the tele02 primer pair (*Taberlet et al., 2018*), which has excellent resolution for Central European fish species (*Macher et al., 2023b*). A total of 176 first-step PCRs were performed per primer. The initial 19 samples were analysed along with one PCR negative control (containing PCR-grade water), one extraction negative control, and the field blank (*i.e.,* 22 PCR reactions). All samples were run in replicates (*i.e.,* 44 PCR reactions) and separately for each target nucleic acid (*i.e.,* 88 PCR reactions) for the two primer pairs (*i.e.,* 176 PCR reactions in total).

The 1st-step PCR volume was 25 µL, consisting of 4 µL of PCR-grade water, 12.5 µL of Multiplex Mastermix (Qiagen Multiplex PCR Plus Kit; Qiagen, Hilden, Germany), 2.5 µL of CoralLoad dye, 0.5 µL of forward primer (fwhF2 /tele02_f, 10 µM), 0.5 µL of reverse primer (fwhR2n/tele02_r, 10 µM), and 5 µL of DNA/RNA template. For both primers, touchdown PCRs with the following settings were conducted: denaturation at 95 °C for 5 min, 10 touchdown cycles with 95 °C for 30 s, annealing for 90 s (fwh = 68−1 °C; tele02 = 62−1 °C), and 72 °C for 90 s. After the touchdown cycles, 25 cycles with optimal annealing temperature were performed (fwh = 58 °C; tele02 = 52 °C). The final elongation was carried out at 68 °C for 10 min.

For the 2nd-step PCR, a universal tagging primer set was used (*Buchner, Haase & Leese, 2021*). A total of 176 2nd-step PCRs were conducted. The PCR mix per sample contained 1.8 µL of PCR-grade water, 7.5 µL of Multiplex Mix, 1.5 µL of CoralLoad dye, 1.2 µL of combined sample-specific tagging primer (5 µM) and 3 µL of 1st-step PCR product. PCR conditions were 95 °C for 5 min, followed by 10 cycles at 95 °C for 30 s and 72 °C for 120 s. The final elongation was carried out at 68 °C for 10 min.

An additional 1st-step PCR using the tele02 primer pair, with the same PCR conditions described above, was conducted using ezDNAse-treated but not reverse-transcribed samples. Thus, it was possible to verify that only cDNA was amplified in the RNA fraction and all DNA was digested during the ezDNAse treatment. The PCR products were visualized on a 1% agarose gel and did not produce bands.

Following the 2nd-step PCR, the PCR products were visualized on a 1% agarose gel to evaluate the amplification success. The samples were subsequently normalized and size-selected (to remove primer dimers) using a magnetic bead protocol (*Buchner, 2022b*). Then, the normalized samples using fwh and tele02 for both fractions were pooled into one library. After library-pooling, the samples were concentrated using a NucleoSpin Gel and PCR Clean-up Kit (Macherey Nagel, Düren, Germany) following the manufacturer's protocol. The final elution volume of the library was 30 µL. The samples were then analysed using a Fragment Analyzer (High Sensitivity NGS Fragment Analysis Kit; Advanced Analytical, Ankeny, IA, USA) to check for primer dimer and co-amplification and to quantify the DNA concentration of the library. The final library was sequenced on a HiSeq X platform using the 150 bp PE Kit at Macrogen (Seoul, Republic of Korea).

## Bioinformatics

Raw reads were received as demultiplexed fastq files. All samples were processed with the APSCALE-GUI pipeline v1.2.0 (*Buchner, Macher & Leese, 2022*), which is based on VSEARCH (*Rognes et al., 2016*) and cutadapt (*Martin, 2011*). Initially, paired-end reads were merged. Then, the dataset was split according to the two markers used. Next, the specific primers were removed (tele02 and fwh) and the samples were separately processed as the tele02 dataset and fwh dataset. Default settings were used (MaxEE = 1), with different length filtering thresholds (fwh = 195–215 bp, and tele02 = 157–177 bp). OTUs were clustered with a 97% percentage similarity threshold, which is the default clustering threshold for the fwh primer (*Vamos, Elbrecht & Leese, 2017*) and has proven to provide good species estimates for the tele02 primer (*Macher et al., 2023b*).

For the fwh dataset, taxon assignments were obtained using BOLDigger (*Buchner & Leese, 2020*) by searches against the BOLDsystems COI database (http://www.boldsystems.org). The resulting taxonomy table was filtered using the 'JAMP filtering' option (98%: species level, 95%: genus level, 90%: family level, 85%: order level, <85%: class level).

For the tele02 dataset, taxon assignments were obtained using the integrated blast+ tool in APSCALE-GUI by blastn searches (somewhat similar sequences) against the midori2 database (GB249 srRNA BLAST; *Leray, Knowlton & Machida, 2022*). The resulting taxonomy table was filtered using the following thresholds: 97% for the species level, 95% for the genus level, 90% for the family level, 85% for the order level, and <85% for the class level.

For both datasets, the taxonomy and read tables were converted to TaXon tables (Tables S1 and S2) for downstream analyses using TaxonTableTools v1.5.0 (*Macher, Beermann & Leese, 2021*). Initially, PCR replicates were merged, and only operational taxonomic units (OTUs) present in both PCR replicates were kept. To account for potential contamination, a strict read filter was applied where the sum of reads per OTU in negative controls was subtracted from the reads per OTU for each sample. The datasets were then filtered by taxonomic groups and then normalized (to the sample with the fewest reads). For the fwh dataset, only OTUs assigned to invertebrates (*i.e.,* Annelida, Arthropoda, Bryozoa, Cnidaria, Mollusca, Nematoda, Nemertea, Platyhelminthes, Porifera and Rotifera) with a similarity of ≥85% were retained. The tele02 dataset was filtered for OTUs assigned to fish (Actinopteri

and Hyperoartia), birds (Aves), and mammals (Mammalia) and only hits with a similarity of ≥85% were kept. These two TaXon tables were used for all downstream analyses (Tables S3 and S4). Additionally, the invertebrate TaXon table was filtered according to the official German freshwater taxon list (Perlodes, http://www.gewaesser-bewertung-berechnung.de) that is used for the Water Framework Directive assessment. Thus, the number of indicator taxa was investigated.

## Statistical analysis

For both datasets the species richness per sample was calculated and visualized as box plots per molecule and scatter plots sorted in sampling order (*i.e.,* time point). A Wilcoxon test was performed to test for differences in the average species richness between eRNA and eDNA metabarcoding, using scipy 1.11.1 (*Virtanen et al., 2020*). Additionally, Jaccard distances were calculated for the eRNA and eDNA fraction of each sample, using scipy, and plotted as a secondary $y$-axis. Subsequently, Jaccard distances were calculated for all possible combinations within the eRNA and eDNA datasets, using the time difference between compared samples as the $x$-axis. Results were visualized in separate scatter plots for both eRNA and eDNA. To test for the effect of time difference on the Jaccard distance, Spearman correlations were calculated, using scipy (Figs. 2 and 3).

Relative species detection rates for eRNA and eDNA were calculated as the percentage of samples with a positive species detection (sample occupancy) and plotted to display the deviation from an expected 1:1 relationship (Figs. 4 and 5).

To evaluate the extent to which eRNA and eDNA data reflect the local fish community at the study site, we used electrofishing data from two restoration monitoring campaigns conducted in the mouth of the river Lippe in 2016 and 2019 (the study site) as well as electrofishing data collected during the regulatory WFD fish monitoring from 2015 to 2019 from the river Lippe mouth and its upstream area (up to 18 km from the mouth). To determine riverine habitat preferences, we calculated the relative occurrence of each species at downstream (wide restored river mouth including still waters, gravel, and sand banks) and upstream sites (unnatural, straightened, and constrained river, high flow velocity and deep waters). By subtracting the occurrence proportions at downstream sites from those of upstream sites, we derived an estimate of the likelihood of occurrence. Preference values ranged from −100% (indicating the strongest downstream preference) to 100% (representing the highest degree of upstream preference). Based on the assumption that eRNA is less stable than eDNA and will thus provide a more local signal, we evaluated the relationship of the differences between eRNA and eDNA detection rates and between numbers of downstream and upstream specimens. Here, we predict the following general patterns: (i) a stronger eRNA signal ratio will be positively associated with a higher likelihood of occurrence in downstream habitats, (ii) a stronger eDNA signal ratio will be related to a preference for upstream habitats, and (iii) a stronger eRNA signal ratio will not be correlated with preferences for upstream habitats (see boxes in Fig. 6). Therefore, the data were tested for normal distribution using the Shapiro–Wilk and Kolmogorov–Smirnov test, using scipy. Since data were not normally distributed a generalized linear model (GLM) with a Gaussian family and an identity link function was fitted to the data, using scipy. Here

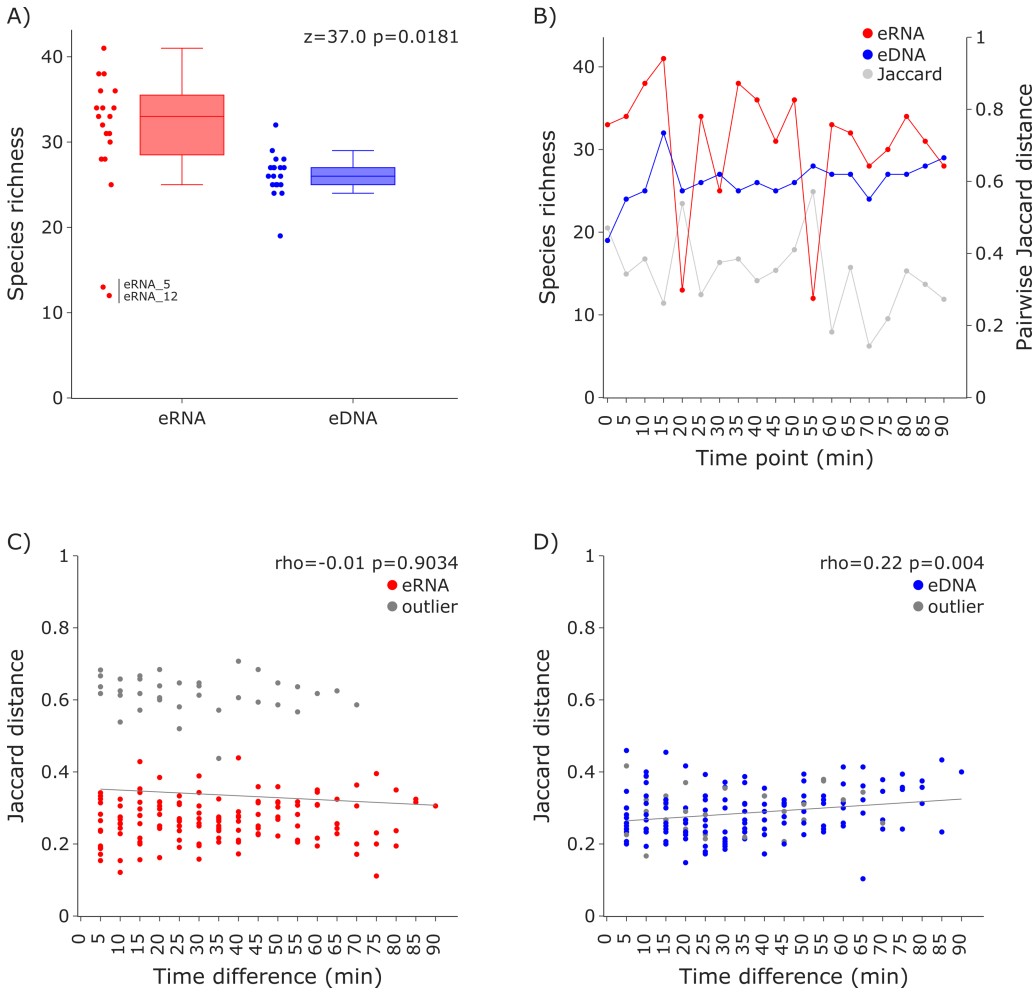

**Figure 2** **Species richness and Jaccard distances over time for the 12S vertebrate datasets.** (A) The eRNA dataset showed significantly higher richness (Wilcoxon $z = 37.0$, $p = 0.0181$). The species richness inferred from the eRNA and eDNA dataset across the 90 min sampling interval is shown in (B). Jaccard distances between all sample pairs of the eRNA (C) and eDNA datasets (D) were calculated. Time differences between sample pairs were calculated and used as $x$-axis values. No trend was observed for the eRNA data, both including (rho $= -0.01$, $p = 0.9034$) and excluding outliers (rho $= 0.1$, $p = 0.2628$), while a slight positive trend for an increase of Jaccard distance with increased time differences between the eDNA samples was observed (rho $= 0.22$, $p = 0.004$). The samples "eRNA_5" and "eRNA_12" and their respective eDNA counterparts are highlighted as outliers (grey) in both the eRNA (C) and eDNA dataset (D).

we chose "eRNA-eDNA detection probability" as the dependent (response) variable and the "change in downstream-upstream specimen numbers" as the independent (predictor) variable. The goodness of fit was evaluated by the pseudo-$R^2$ value.

Rarefaction curves were calculated using TaxonTableTools to investigate the increase in species richness with increased sampling effort. Each draw was repeated 1,000 times and the average number of species was calculated (Fig. S1).

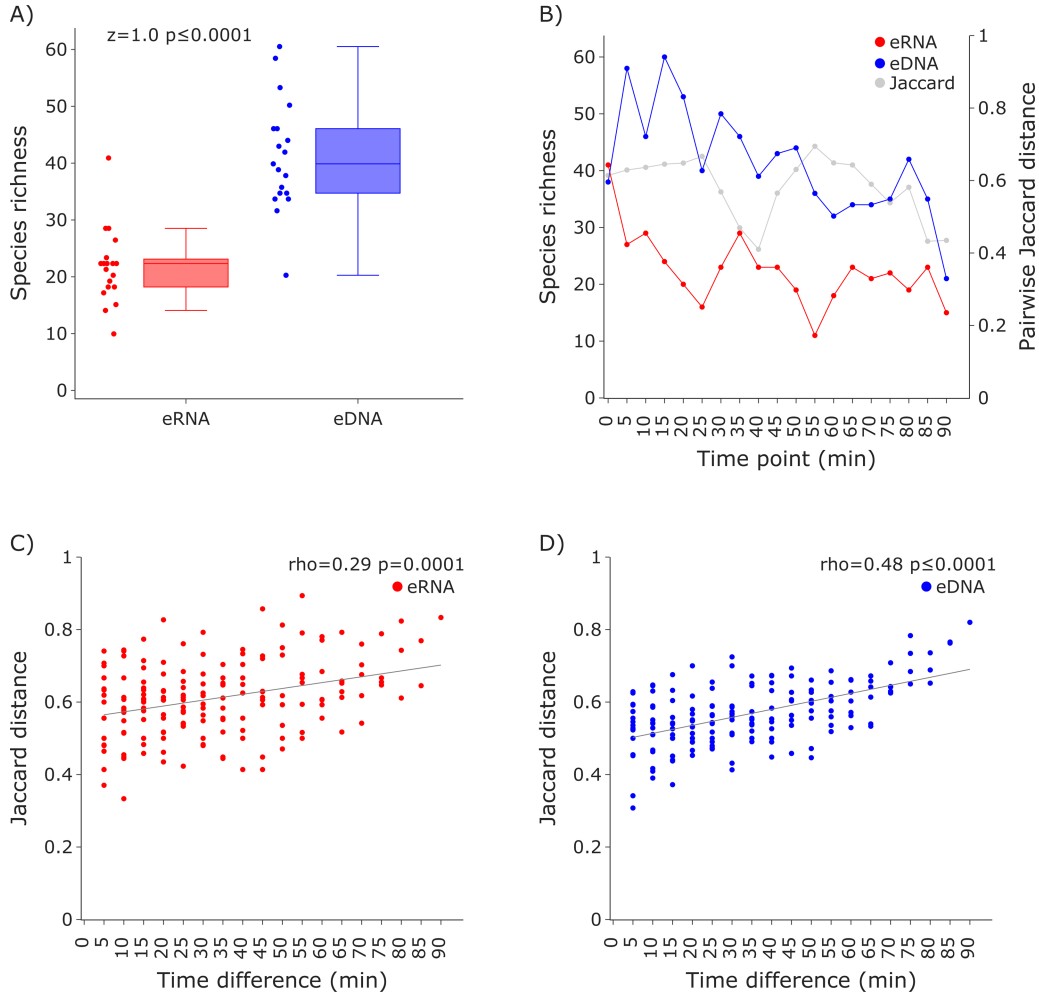

**Figure 3** **Species richness and Jaccard distances over time for the COI invertebrate datasets.** The eDNA dataset showed significantly higher richness (Wilcoxon $z = 1.0$, $p \leq 0.0001$) are depicted in (A). The species richness inferred from the eRNA and eDNA dataset across the 90 min sampling interval is shown in (B). Jaccard distances between all sample pairs of the eRNA (C) and eDNA datasets (D) were calculated. Time differences between sample pairs were calculated and used as $x$-axis values. Jaccard distances significantly increased with time difference for both the eRNA ($rho = 0.29$, $p = 0.0001$) and eDNA dataset ($rho = 0.48$, $p \leq 0.0001$).

Additionally, the eDNA and eRNA samples for both vertebrates and invertebrates were compared per sample (*i.e.,* shared and exclusive species and Jaccard dissimilarity; Figs. S2A and S2B) and as whole datasets using Venn diagrams (Figs. S2C and 2D), both using TaxonTableTools. To investigate the proportion of low-read-abundance species, the relative number of shared and exclusive species (for eRNA and eDNA) for five categories (read abundance >= 10%, <10% to >= 5%, <5 to >= 1%, <1% to >= 0.1%, and <0.1%) were calculated (Fig. S2). Lastly, Principal coordinate ordination analyses (PCoA) based on Jaccard distances were performed using TaxonTableTools (Fig. S4). The 15 most representative species (*i.e.,* species with highest coefficient values) were calculated using a

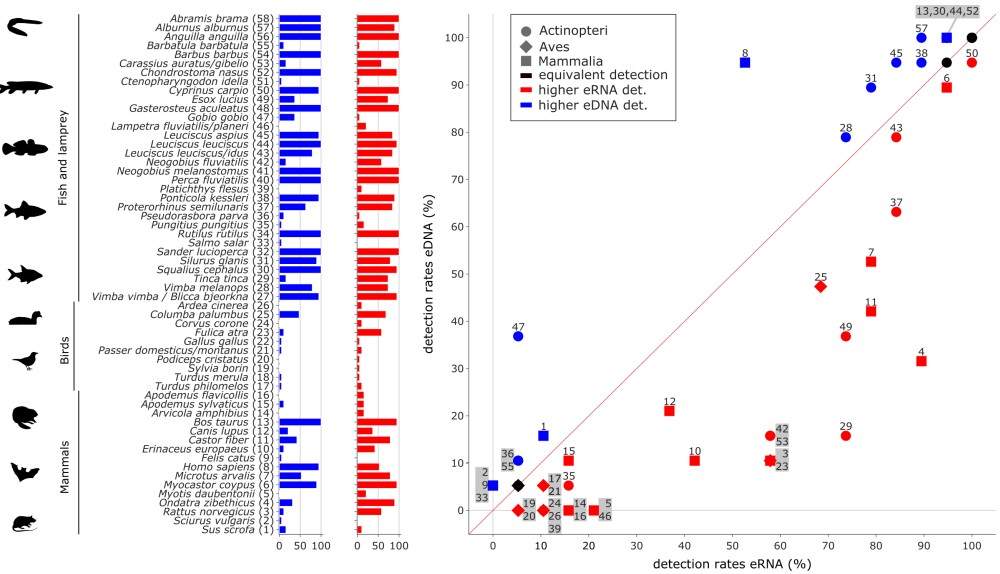

**Figure 4** **Comparison of eDNA (blue) and eRNA (red) detection rates for vertebrate species.** Overall, a significant correlation between the detection rates of both eRNA and eDNA metabarcoding was observed (rho = 0.87, $p \leq 0.0001$). However, more species were observed with higher rates using eRNA (29 species) than using eDNA metabarcoding (17 species; Wilcoxon $z = 284$, $p = 0.005$).

logistic regression model and are displayed as vectors in the PCoA (vector scaling factor 0.3).

## RESULTS

### Vertebrates

In total, 245,217,826 quality-filtered reads were clustered into 1,093 OTUs for the tele02 dataset. After replicate merging, 242,477,772 reads remained. Of these, 235,690,432 reads (97.2%) were assigned to samples and 6,787,340 (2.8%) to field blanks and negative controls. The final dataset consisted of 42,068,318 reads assigned to 746 vertebrate OTUs (after the subtraction of negative controls, removal of non-target OTUs and read normalization). On average, we detected 487.2 OTUs per sample. However, the two samples 'eRNA_5' and 'eRNA_12' showed significantly fewer OTUs (105 and 193, respectively) despite similar read counts and were thus marked as outliers.

In total, we detected 31 fish (Actinopteri), one lamprey (Hyperoartia), 10 bird (Aves), and 16 mammalian (Mammalia) species. Of these species, three were found only in the eDNA dataset, 46 were shared, and nine occurred only in the eRNA dataset (Fig. S2C).

Species richness (Fig. 2A) was higher in the eRNA dataset (average = 30.89) than in the eDNA dataset (average = 26.16; Fig. 2A). Most species detected were assigned to the class Actinopteri (median of 21 for eDNA and 23 for eRNA), followed by Mammalia (five for eDNA and seven for eRNA), and Aves (zero for eDNA and two for eRNA). Lampreys (Hyperoartia) were detected in only four of the 19 eRNA samples, leading to a median of zero. Species richness detected over the 1.5 h sampling interval varied stronger for the eRNA

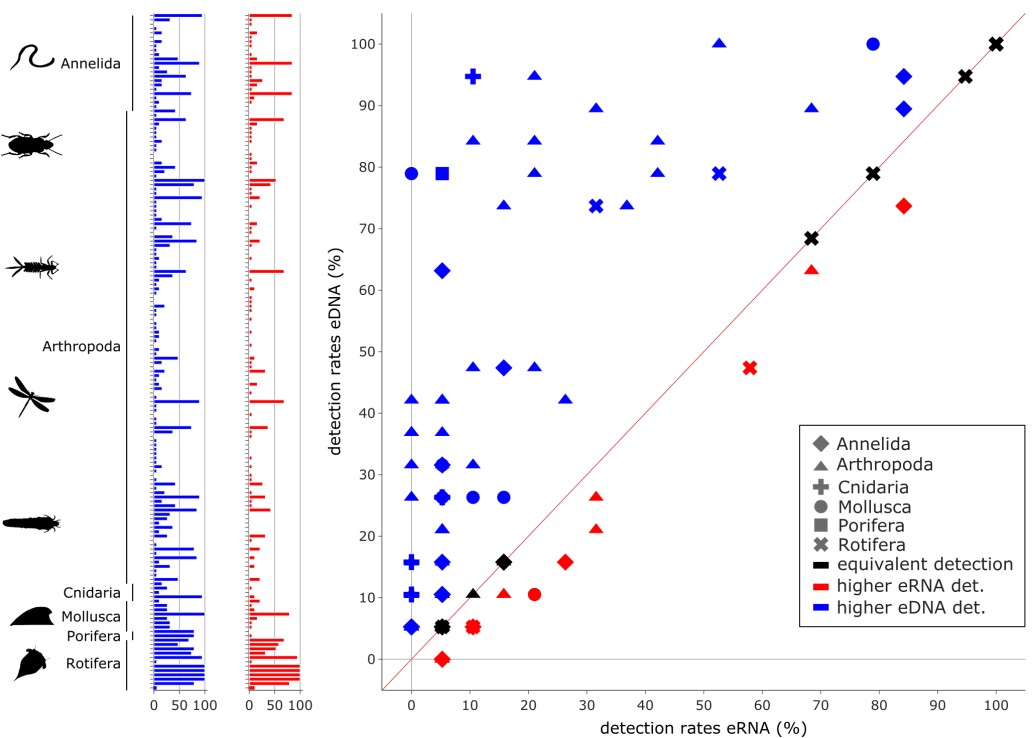

**Figure 5  Comparison of environmental DNA (blue) and RNA (red) detection rates for invertebrate species.** Overall, a significant correlation between the detection rates of both eRNA and eDNA metabarcoding was observed (rho = 0.59, $p \leq 0.0001$). However, eDNA metabarcoding detected a significantly higher number of species, observing 105 species, compared to eRNA metabarcoding, which detected 27 species (Wilcoxon $z = 1100.0$, $p \leq 0.0001$).

samples (SD ± 7.34) than for the eDNA samples (± 2.5). Community differences between sampling time points were calculated using Jaccard distances. For eRNA and eDNA, the pairwise Jaccard distances varied between 0.14 and 0.57 (average = 0.34; Fig. 2B, secondary *y*-axis). For the eRNA dataset inferred community composition did not increase with increasing time interval between sampling time points, both when including (rho = −0.01, $p = 0.9034$) and excluding the two outlier samples (rho = 0.1, $p = 0.2628$; Fig. 2C). For the eDNA dataset, Jaccard distances slightly increased with time highlighting a small but significant increase in community distance with increasing time between sampling (rho = 0.22, $p = 0.004$; Fig. 2D).

Most species with a higher detection rate (*i.e.,* the number of replicates where the species was detected) using eRNA metabarcoding (Fig. 4) were birds (eRNA = 8 species, eDNA = 0 species), and mammals (eRNA = 11, eDNA = 5). Interestingly, for fish species, the detection rate was higher using eDNA metabarcoding (eRNA = 10, eDNA = 12). Species exclusively detected by either eDNA or eRNA metabarcoding always had few reads, *i.e.,* <0.1% of all reads (Fig. S2A).

The eDNA and eRNA samples formed distinct groups when investigating beta diversity and including all samples (Jaccard dissimilarity, Anosim $R = 0.39$, $p = 0.001$). This

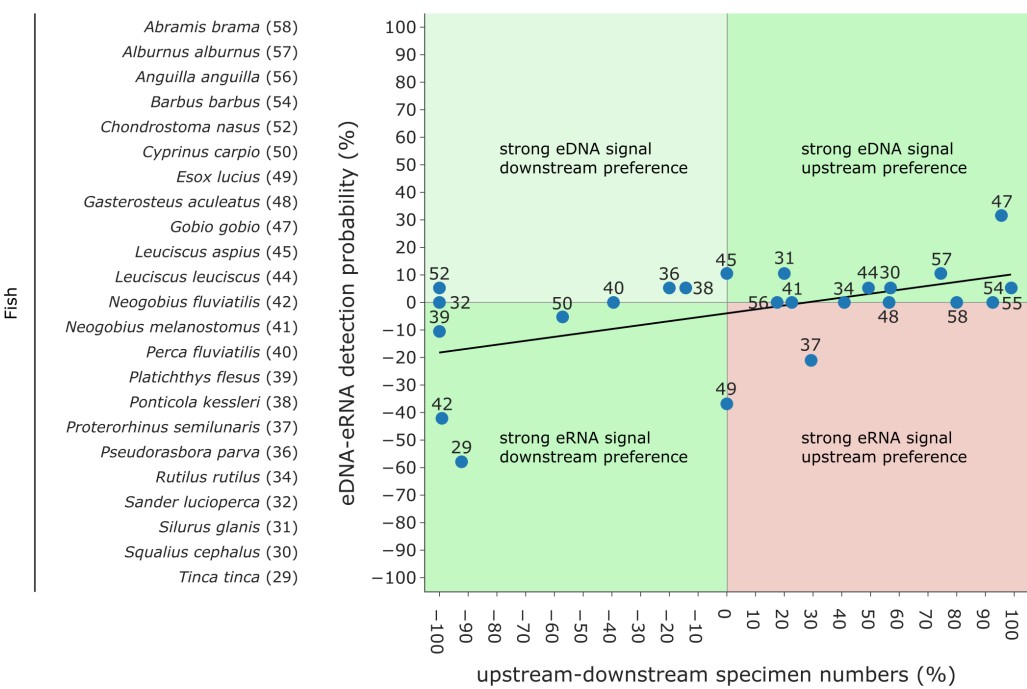

**Figure 6 Comparison of eDNA and eRNA fish detection rates to downstream & upstream fish specimen numbers obtained by electrofishing campaigns (data from 2018 and 2019).** Species are labelled according to Fig. 4. While strong "eRNA signal & downstream preferences" and "strong eDNA signals & upstream preferences" are expected (green), "strong eRNA signals & upstream preferences" disagree with the hypothesis that eRNA reveals a more local signal (red). Lastly, "strong eDNA signals and downstream preferences" can occur (light green). A generalized linear model (GLM) with a Gaussian family and an identity link function was fitted to the data (Pseudo- $R^2 = 0.275$).

difference between sample types was greater when removing the two outlier samples 'eRNA_5' and 'eRNA_12' (Anosim $R = 0.5$, $p = 0.001$; Fig. S3A). Here, the most representative species according to the logistic regression analysis (*i.e.,* species with the highest coefficient values) were *Ondatra zibethicus*, *Tinca tinca*, *Fulica atra*, *Castor fiber*, *Columba palumbus* and *Microtus arvalis*. Most variance in the PCoA was explained by species with higher detection rates in the eRNA data. Overall, eDNA and eRNA data showed high numbers of shared species (65% overall species; Fig. S1A).

Traditional fish monitoring data based on 3,929 fish specimens caught near the river Lippe mouth and 6,885 specimens caught further upstream (Table S7) confirmed all 24 species detected with genetic methods. The traditional fish data collected through various years and at multiple locations detected 12 additional species. From the list of 24 species detected with genetic methods, three were only detected in the river mouth, six were more common downstream, two were detected in equal numbers down- and upstream, and the remaining 13 were more frequent in upstream regions. Based on genetic data, six species had a higher eRNA detection rate, eight were detected by the two methods equally, and the remaining ten species showed higher eDNA detection rates. When comparing the eRNA:eDNA detection rate with the downstream to upstream specimen numbers, we

observed two species (*Tinca tinca* and *Neogobius fluviatilis*) with a strong eRNA signal ratio and downstream preference (Fig. 6, lower left square), while *Gobio gobio* showed a stronger eDNA signal ratio and upstream preference (Fig. 6, upper right square). *Proterorhinus semilunaris* was the only species with a strong eRNA signal ratio and a presumable upstream preference (Fig. 6, lower right square). The remaining species did not show preferential occurrences in up- or downstream sections and showed similar detection rates. The conducted Shapiro–Wilk test ($W = 0.8$, $p = 0.0003$) and Kolmogorov–Smirnov test ($D = 0.42$, $p = 0.0003$) suggested non-normal distributed data. Thus, the data were analysed with a GLM and a moderate fit was observed (pseudo-$R^2 = 0.27$).

The rarefaction curve analyses of the vertebrate eRNA and eDNA datasets showed a high average number of species detected for both eRNA (30.8 average species) and eDNA (26.2). However, with increasing sampling effort more species were detected. While eRNA required four, 10, and 19 samples to reach 75%, 90% and 100% of detected vertebrate species (Fig. S1A), eDNA required five, 12, and 19 samples (Fig. S1B).

## Invertebrates

In total, 128,778,195 quality-filtered reads were clustered into 5,036 OTUs. After replicate merging, 128,024,400 reads remained. Of these, 128,024,047 reads (99.9%) were assigned to samples and 353 reads (0.0003%) to field blanks and negative controls. The final dataset consisted of 4,286,096 reads assigned to 797 invertebrate OTUs (after the subtraction of negative controls, removal of non-target OTUs and read normalization). A high number of reads (126,575,041) were assigned to bycatch taxa (mainly Bacillariophyceae). We detected 156 invertebrate species, including 15 only found in the eRNA dataset, 86 shared, and 55 exclusively detected by analysing eDNA (Fig. S1D). After filtering for indicator taxa (according to the German Invertebrate Operational Taxa list for Water Framework Directive monitoring), 17 taxa remained for the eRNA dataset and 19 for the eDNA dataset (Tables S8 and S9).

Species richness was higher in the eDNA dataset than in the eRNA dataset (Fig. 3A). Most species were assigned to the phylum Arthropoda (median eDNA = 21 and median eRNA = 8), followed by Rotifera (9 and 9), Annelida (6 and 4), Mollusca (3 and 1), Cnidaria (1 and 0), and Porifera (1 and 0). The species richness over time varied stronger for the eRNA samples (SD ± 9.25) than for the eDNA samples (± 6.22). Species richness ranged from 11 to 41 species (average = 22.42) for the eRNA fraction and 21 to 60 species (average = 41.37) for the eDNA fraction (Fig. 3B, primary $y$-axis). Jaccard distances between the eRNA and eDNA fractions of each sample varied between 0.41 and 0.69 (average = 0.58; Fig. 3B, secondary $y$-axis). Generally, the inferred community composition changed significantly with sampling time between sample pairs (eRNA: rho = 0.29, $p = 0.0001$; eDNA: rho = 0.48), suggesting that the signal for invertebrates varies stronger across time than for vertebrates.

Both the eDNA and eRNA samples formed distinct groups when investigating species (PCoA, Jaccard index, Anosim $R = 0.55$, $p = 0.001$; Fig. S3B). Here, the most representative species according to the logistic regression analysis (*i.e.,* species with the highest coefficients) were *Tipula couckei*, *Cladotanytarsus mancus*, *Ephydatia fluviatilis*, *Hydra vulgaris* and

*Cryptochironomus rostratus*. Most variance in the PCoA was explained by species with higher detection rates in the eDNA data.

The eDNA and eRNA datasets were dominated by reads assigned to the rotiferan order Ploima, which accounted for 94.6% and 88.7% of reads in the eRNA and eDNA dataset, respectively, while significantly fewer reads were assigned to all remaining invertebrate orders. However, the difference in species composition between eDNA and eRNA datasets was only attributed to species with low read abundances of <0.1% (Fig. S2B).

In total, 105 species were detected more often by eDNA metabarcoding than by eRNA metabarcoding, while 27 species were detected more often by eRNA metabarcoding. In total, 24 species were equally detectable by either method (Fig. 5; Table S6). The phyla with the highest proportions of species with a higher eDNA detection rate were Cnidaria (100%, 4/4 species), Porifera (100%, 1/1), Mollusca (87.5%, 7/8), Arthropoda (71.6%, 78/109) and Annelida (59%, 13/22). Rotifer species were detected equally well by eRNA and eDNA (66.7%, 8/12). No phylum had higher detection rates when analysing eRNA.

The rarefaction curve analyses of the vertebrate eRNA and eDNA datasets showed a comparably low average number of species detected for both eRNA (22.4 average species) and eDNA (41.3), both accounting for less than 50% of all species detected in the respective dataset. Here, with increasing sampling effort more species were detected. To reach 50% (eRNA: five samples, eDNA: four), 75% (11, nine), 90% (16, 15) and 100% (19, 19) of all detected invertebrate species, a similar number of samples were required for both approaches (Figs. S1C and S1D).

# DISCUSSION

## Species richness and detection rates

Our study found consistent species detections for both eDNA and eRNA metabarcoding in vertebrates, aligning with previous studies on fish species in lakes (*Littlefair, Rennie & Cristescu, 2021*) and rivers (*Miyata et al., 2021*). However, we noticed differences in species richness and detection rates between groups. Environmental RNA metabarcoding detected a higher number of mammal and bird species, contributing to its overall higher species numbers. In contrast, eDNA metabarcoding had slightly higher detection rates for fish but failed to detect lampreys, which were only found with eRNA metabarcoding. Interestingly, two samples from the eRNA metabarcoding approach were outliers, with significantly fewer detected species compared to other eRNA samples. These outlier samples did not show signs of lower RNA amounts or suffer from lower sequencing depth. We hypothesize that the additional step of reverse transcription into cDNA may have negatively impacted these samples, suggesting that eRNA metabarcoding could be more susceptible to outliers than eDNA metabarcoding. Further investigations of potential sources of bias in the eRNA metabarcoding workflow are recommended to address this issue and if needed, minimize these in future studies. Due to the additional required steps, such as cDNA synthesis that can introduce extra noise (*Verwilt, Mestdagh & Vandesompele, 2023*), eRNA metabarcoding is generally more prone to bias or outliers. While the exact reason for the higher eRNA detection rates for vertebrates is unclear, we speculate that it could be due to a higher initial

concentration of mitochondrial ribosomal RNA than mitochondrial DNA, as observed for 16S rRNA (*Marshall, Vanderploeg & Chaganti, 2021*). We assume that this also applies to the 12S marker used in our study, as 16S and 12S rRNA are both components of the mitochondrial ribosome.

Contrary to vertebrate species richness, we found lower invertebrate richness for eRNA than for eDNA. Specifically, the eDNA dataset showed higher average species numbers for Annelida, Arthropoda, Cnidaria, Mollusca, and Porifera. This is consistent with the findings of *Miyata et al. (2022)* who also reported more invertebrate species detected by eDNA metabarcoding than by eRNA metabarcoding.

## Recommendations for optimal sampling efforts in biodiversity monitoring

Given the observed differences in species detection between eDNA and eRNA metabarcoding, particularly for invertebrates, it becomes crucial to consider these factors when planning biodiversity monitoring efforts. In flowing waters, the temporal dynamics of eDNA signals have been noted to change (*Macher & Leese, 2017*; *Sales et al., 2021*; *Troth et al., 2021*), but the extent of non-homogeneous signal distribution in eRNA metabarcoding analyses remains unexplored. For our vertebrate data, we detected only minimal changes in inferred community composition between samples for both eDNA and eRNA across the 90-minute sampling interval. These results suggest that both eRNA and eDNA molecules of vertebrates are present in similar concentrations across time. Several eDNA-based biodiversity monitoring studies already emphasized that to recover also rare species replication is crucial (*Cantera et al., 2019*; *Beentjes et al., 2019*; *Macher et al., 2021*). Generally, while the absolute species richness differed, the increase in detected species (*i.e.,* the slope of the rarefaction curve) was similar between eRNA and eDNA. Overall, these results align with study by *Macher et al. (2021)*, which suggested 8–10 field replicates (of 1 L samples, instead of 3 L as used here) as a good trade-off between sampling effort *versus* species detection for eDNA metabarcoding analysis of a similarly sized river.

Similar to the vertebrate dataset, we observed consistent patterns for the invertebrate dataset. However, here the overall detected species richness was higher for the eDNA samples. Nevertheless, we observed a similar decrease of species richness across the 90-minute sampling interval time for both eDNA and eRNA. Why the species richness continuously decreased remains unclear. Consequently, a significant increase in Jaccard distance with time difference was observed for both the eRNA and eDNA invertebrate datasets. Generally, Jaccard distances were high for both invertebrate datasets, suggesting similar non-homogenous distribution of eRNA and eDNA invertebrate signals in the water. This was also reflected in the rarefaction curves of the invertebrate datasets. While the eDNA dataset contained more species, the relative increase was similar for both datasets. These results, again, highlight that biodiversity monitoring campaigns should ideally include several field replicates to gain a more complete picture of the investigated site, both for eDNA and eRNA metabarcoding.

Our results stress the similarities of eRNA and eDNA template distribution in streams, yet differences also exist among taxonomic groups, as demonstrated for the vertebrate and

invertebrate datasets. Given the comparable behaviour of eRNA and eDNA signals within species, insights derived from sampling designs in eDNA studies, including the necessary number of samples, can likely be applied to eRNA investigations. It should be noted that we sampled water only at a single site. In other stream types and under different environmental conditions eRNA and eDNA signals might behave differently. Future studies should thus conduct sampling over different time periods and various stream types to provide an even more comprehensive evaluation of species richness and distribution patterns.

## Marker choice in eRNA metabarcoding

Compared to eDNA metabarcoding, marker selection might be more relevant for eRNA metabarcoding analyses due to differences in the function of the transcribed RNA molecules. We used two commonly employed primer pairs for eDNA-based fish (tele02 primer pair) and invertebrate (fwh primer pair) biodiversity assessment, not considering potential marker region variations. In our investigation, the eRNA vertebrate dataset, derived from a 12S rRNA marker, displayed higher species richness and detection rates. Conversely, the eRNA invertebrate dataset derived from a COI mRNA marker had lower species richness and detection rates. While both are mitochondrial markers and should be equally abundant in their eDNA form, their transcribed forms differ significantly. Ribosomal genes like 12S rRNA are transcribed into rRNA, forming stable ribosome complexes, whereas mRNA, such as COI mRNA, is single-stranded and less stable due to its susceptibility to endonuclease digestion. These crucial differences between markers necessitate consideration in eRNA-based studies.

*Marshall, Vanderploeg & Chaganti (2021)* illustrated these distinctions under controlled conditions. They observed higher initial concentrations but lower half-lives of 16S rRNA and COI mRNA compared to 16S rDNA and COI DNA in dreissenid mussels. Notably, 12S rRNA and 16S rRNA, forming the mitochondrial ribosome, might behave similarly in their eRNA form. The high concentrations of 16S rRNA resulted in elevated detection rates even after 240 h post-dreissenid mussel removal, aligning with the higher vertebrate species richness and detection rates in our eRNA dataset. Conversely, other studies reported higher eRNA decay rates for the mitochondrial COI marker (*Kagzi et al., 2022*), nuclear beta-2-microglobulin for zooplankton, and mitochondrial cytochrome b for zebrafish (*Jo et al., 2022*), all of which are translated to mRNA. These findings correspond with the differences in species richness and detection rates between the rRNA and mRNA markers, used for vertebrates and invertebrates.

However, our study did not directly compare rRNA and mRNA markers for vertebrates and invertebrates individually, limiting possible conclusions and rendering these results preliminary. Future studies should explore multiple markers per organism group from field samples to validate these assumptions. Nonetheless, acknowledging the disparities among eRNA markers is crucial when planning eRNA-based studies, given the variability within eRNA, which was not required for eDNA-based approaches.

Nevertheless, invertebrate and specifically benthic macroinvertebrate diversity was underestimated in both datasets, particularly when compared to a parallel eDNA metabarcoding study conducted at the same site, which found between 60 and 146

aquatic invertebrate species over the course of one year (*Macher, 2023*). For example, after filtering for indicator taxa (according to the WFD assessment), only 19 taxa remained for the eDNA dataset and 17 for the eRNA dataset. Most discarded taxa were either not relevant for ecological status class assessments (*e.g.*, rotifers) or were terrestrial. Three key factors may explain these low numbers of detected species. First, the water samples were collected during a high tide, which could have diluted the stream water and thus, reduce macroinvertebrate eDNA and eRNA, *e.g.*, as reported for fish (*Sales et al., 2021*). Second, both eDNA and eRNA metabarcoding approaches were strongly affected by the high co-amplification rates of diatoms (Bacillariophyta; 71% of all reads), reducing the sequencing depth for target organisms drastically. Here, both datasets were similarly affected. Third, rotifers accounted for more than 90% of the remaining reads. Rotifers are part of the freshwater zooplankton and therefore, are highly abundant in the water column, likely contributing more eDNA and eRNA in the water column compared to ground-dwelling macroinvertebrates. Thus, only few reads were assigned to many relevant taxa in the context of biodiversity monitoring, such as the EPT taxa Ephemeroptera (two reads) and Trichoptera (nine reads) and Plecoptera (no reads). The fwh primer was originally designed to increase target specificity for macroinvertebrates using bulk samples (*Vamos, Elbrecht & Leese, 2017*) and has generated robust taxon lists in various studies (*Buchner et al., 2023*; *Newton et al., 2023*). On the other hand, it can amplify large proportions of non-target taxa when using environmental and not bulk samples (*Leese et al., 2021*). Here, the usage of more target-specific primers, such as the fwhF2/EPTDr2n primer pair (*Leese et al., 2021*), developed to improve freshwater macroinvertebrate detection, could improve assay sensitivity (*Brantschen et al., 2021*). Additionally, future studies should consider potential distinct decay rates and evaluate signal persistence not only when comparing the same marker gene but targeting different regions of different lengths. Specifically, we recommend that future eRNA metabarcoding studies should compare the performance of primer pairs targeting different marker genes (*i.e.,* nuclear and mitochondrial) to test for differences in decay rates depending on the target molecule (mRNA *vs.* ribosomal RNA) by comparing the success of different amplicon lengths of the same marker gene. Here, many aspects of the eRNA metabarcoding workflow are yet not sufficiently understood.

## Evaluation of signal locality

The 'ecology' of eRNA suggests that it has the potential to capture a more localized signal, as it is released by living organisms and exhibits more rapid degradation than that of eDNA (*Cristescu, 2019*; *Littlefair, Rennie & Cristescu, 2021*). For example, eRNA has a 4–5 h shorter half-life compared to eDNA (*Marshall, Vanderploeg & Chaganti, 2021*). Therefore, when comparing preferences of species for downstream and upstream habitats (*e.g.*, derived from traditional fish monitoring results of the area conducted over the past years) based on eRNA and eDNA, we expected a stronger eRNA:eDNA signal ratio corresponding to a downstream preference and a stronger eDNA signal ratio corresponding to an upstream preference for fish species (see boxes in Fig. 6). However, our GLM analyses only found a weak positive correlation. Overall, most species were detected equally well with eDNA and eRNA. Two species, the tench (*Tinca tinca*) and the monkey goby (*Neogobius fluviatilis*)

showed a downstream preference with a higher signal ratio for eRNA metabarcoding, while the gudgeon (*Gobio gobio*) exhibited an upstream preference with higher signal ratio for eDNA metabarcoding. The interpretation of the results proves difficult since most fish species are found in both down- and upstream areas. However, tench prefers shallow, calm, and slowly-flowing freshwater environments (*Zarkami, Goethals & Pauw, 2010*), corresponding to the geohydrological characteristics of the river mouth (refer to Fig. 1, "Aue" (German for floodplains)). In this context, the higher detection rate by eRNA metabarcoding may be related to higher metabolic activity of the species (*e.g.*, more specimens) in the river mouth. Therefore, the tench habitat preference, rate of eRNA detection, and traditional monitoring results were consistent. Nevertheless, for two species, detection results did not follow the expected pattern: the northern pike (*Esox lucius*) and western tubenose goby (*Proterorhinus semilunaris*). Both species are likely to inhabit both the river mouth and upstream parts of the river Lippe. Thus, the interpretation of results for these two species remains difficult. Generally, more data from multiple sampling sites will be required to investigate the locality of eRNA signals in a broader context. The presented data may indicate an eRNA to eDNA detection strength gradient along the stream but was limited to a single site.

It is important to note, however, that the traditional fish monitoring data were collected over several years before the collection of water samples. This, in combination with several known sampling biases of both metabarcoding and traditional methods, may have affected comparative analyses. For metabarcoding analyses of water samples, various factors, such as the inhomogeneous distribution of genetic material in the water body, current discharge, seasonal effects, or sampling effort, can affect estimates of species richness (*Hallam et al., 2021*; *Pont et al., 2021*; *Macher et al., 2021*). For traditional monitoring, the size of the sampled water body, flow velocity, turbidity of the water, dead wood or vegetation in the water body, collection effort (*Pusey et al., 1998*; *Rosenberger & Dunham, 2005*; *Bertrand, Gido & Guy, 2006*; *Meyer & High, 2011*), and species-specific habitat preferences (*Polačik et al., 2008*), may result in the over- or underestimation or non-detection of species. While the fish species distributions derived from DNA metabarcoding and traditional restoration and WFD monitoring likely provide a reasonable estimate of the true species distributions, more comprehensive and detailed fish monitoring at the site is necessary to generate closely comparable data. Furthermore, comprehensive species distribution data from the mouth of the river Lippe are not available for mammals or birds, making further comparisons impossible for these groups at this point.

## Application potential of eRNA metabarcoding in biodiversity monitoring

The selection between eDNA and eRNA metabarcoding, as well as the choice of markers, greatly hinges upon the research question. For eDNA metabarcoding, the stability of double-stranded mitochondrial genes in the environment renders them capable of traveling downstream for several kilometers (*Deiner & Altermatt, 2014*; *Shogren et al., 2017*). This approach offers comprehensive estimates of species richness for the entire community. However, eDNA signals might originate from further upstream or from non-active sources

such as carcasses or faecal matter. In contrast, when considering eRNA metabarcoding, the choice of markers significantly influences the signal's characteristics. For instance, mRNA-based approaches offer insights into the active community due to mRNA being single-stranded and only released by active organisms. This specificity potentially leads to a lower observed species richness but offers a clearer picture of the local active community (Fig. S5). Conversely, ribosomal markers, being part of a more stable ribosome complex, may provide higher species richness but potentially offer less locality information due to their stable nature within the ribosome (Fig. S5).

Despite its advantages, the application of eRNA analyses may be constrained by higher costs and labour compared to eDNA metabarcoding. Handling eRNA samples requires immediate freezing or the use of specific preservatives like RNAlater, adding complexity to fieldwork. Additionally, the cost of reverse transcription for RNA to cDNA, essential for subsequent PCR amplification, increases the per-sample cost by approximately 20 USD compared to (e) DNA metabarcoding workflows (*Buchner et al., 2021*). Another potential cause of bias when working with eRNA is the reverse transcription step itself. Like DNA polymerases, reverse transcriptase enzymes exhibit a certain error rate. However, this error rate is added to the natural errors occurring during PCR in any metabarcoding workflow. Nevertheless, the error rates for DNA polymerases and reverse transcriptases are low (*e.g.*, between $5.2 \times 10^{-5}$ and $8.4 \times 10^{-5}$ errors/base; (*Potapov et al., 2018*)) and are unlikely to artificially increase OTU or even species diversity, particularly when a 3% threshold for OTU clustering is used. The subsequent metabarcoding workflow (*i.e.,* 1st-step and 2nd-step PCR, clean-up and sequencing) does not demand additional costs, since cDNA samples can be handled like eDNA samples.

## Outlook

Environmental RNA metabarcoding is a promising new tool for biodiversity studies that could potentially overcome some of the challenges associated with eDNA-based signals. Our investigation demonstrated that using eRNA metabarcoding of 12S rRNA can result in higher species richness and improved detection rates when targeting vertebrates. In contrast, eRNA metabarcoding of COI mRNA aimed at invertebrates resulted in lower species richness and detection rates, although it may be more effective at detecting the local stream community. Hence, the value of eRNA metabarcoding significantly depends on the specific research question. In resource-constrained assessments focused on common fish species within a catchment, eDNA metabarcoding remains the preferable method. However, when the objective is to investigate localized signals and monitor the active community, eRNA metabarcoding can be a valuable alternative. Furthermore, the choice of targeted eRNA markers should be considered when designing eRNA-based studies, since rRNA and mRNA markers may differ in their stability and thus taxon detection probabilities. Here, further studies will need to investigate the effect of primer choice in more detail. In conclusion, eRNA metabarcoding can be worth the extra costs and effort for biomonitoring of streams, when the analysis of local signals and particularly of the active community are of interest.

## ACKNOWLEDGEMENTS

We thank the Emschergenossenschaft und Lippeverband (EGLV) and particularly Gunnar Jacobs for support in organizing the sampling. We thank Nikola Theißen (LANUV) for providing the WFD monitoring data. We thank Marie Brasseur for her input on the statistical analyses.

### Funding

The German Environment Agency is funding the REFOPLAN project GeDNA - (e)DNA-based methods in regulatory context - FKZ 3719 24 2040 in the period 2019—2023. The APC for this article was supported by the Open Access Publication Fund of the University of Duisburg-Essen. The funders had no role in study design, data collection and analysis, decision to publish, or preparation of the manuscript.

### Grant Disclosures

The following grant information was disclosed by the authors:
German Environment Agency: FKZ 3719 24 2040.
Open Access Publication Fund of the University of Duisburg-Essen.

### Competing Interests

The authors declare there are no competing interests.

### Author Contributions

- Till-Hendrik Macher conceived and designed the experiments, performed the experiments, analyzed the data, prepared figures and/or tables, authored or reviewed drafts of the article, and approved the final draft.
- Jens Arle analyzed the data, authored or reviewed drafts of the article, and approved the final draft.
- Arne Beermann analyzed the data, authored or reviewed drafts of the article, and approved the final draft.
- Lina Frank performed the experiments, analyzed the data, authored or reviewed drafts of the article, and approved the final draft.
- Kamil Hupało performed the experiments, analyzed the data, authored or reviewed drafts of the article, and approved the final draft.
- Jan Koschorreck analyzed the data, authored or reviewed drafts of the article, and approved the final draft.
- Robin Schütz conceived and designed the experiments, performed the experiments, analyzed the data, authored or reviewed drafts of the article, and approved the final draft.
- Florian Leese analyzed the data, authored or reviewed drafts of the article, and approved the final draft.

## DNA Deposition

The following information was supplied regarding the deposition of DNA sequences:

The raw data are available at the European Nucleotide Archive: PRJEB72883.
https://www.ebi.ac.uk/ena/browser/view/PRJEB72883.

## Data Availability

The Python scripts used for the statistical analyses are available in the Supplementary File.

## Supplemental Information

Supplemental information for this article can be found online at http://dx.doi.org/10.7717/peerj.18016#supplemental-information.

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
