# Peer review of "Is it worth the extra mile? Comparing environmental DNA and RNA metabarcoding for vertebrate and invertebrate biodiversity surveys in a lowland stream"

_PeerJ, doi:10.7717/peerj.18016_

## Round 0.1 · original submission · Major Revisions

Dear Dr. Macher

The reviewers have commented on your manuscript. You can find attached reports. Based on the comments and suggestions of the expert reviewers, a major revision is needed for your article.

I would like to request you check and correct the manuscript step by step based on the reports.

Sincerely yours

·

Basic reporting

In this manuscript, submitted by Macher et al., a comparison of eDNA vs eRNA metabarcoding is provided aiming at unveiling the diversity patterns for vertebrates around a lowland stream. This comparison is timely and interesting as the use of eRNA is increasing, even though its limitations remain poorly investigated. Please see below a few comments and suggestions, aiming at improving the present manuscript.
I commend the authors for the detailed information provided in the methods section as these would allow for study replicability and will also contribute to future studies.
One interesting methodological choice that could be explored more in depth, refers to the normalization of samples (especially after sequencing). Although this is usually conducted to avoid biases in sequencing depths, it remains controversial as considering that metabarcoding data is usually compositional this step could impact detection rates and therefore, lead to some false negatives in the dataset. It would be beneficial to have a discussion around this included in the manuscript.
One of the limitations of the study refers to the spatial-temporal scale used. Only one site is sampled, over 1.5h and including 19 samples. Although this is mentioned in the discussion, I believe that considering the importance of spatial data for some of the inferences made (e.g., local signals retrieved from eDNA and eRNA and the associated NA ecology), this limitation could be better explored and explained in the discussion, with some perspective included regarding future studies.
The authors suggest that outliers in RNA dataset may be due to issues during reverse transcription into cDNA. More background information is required here for a better explanation of potential challenges during this step.
Another interesting aspect that could be explored in this study refers to the potential effect of fragment size on eDNA/eRNA persistence. The study highlights the remarkable difference found between mitochondrial ribosomal NA vs mitochondrial NA regarding copy numbers and decay rates. It is known that eDNA studies should primarily focus on shorter fragments to increase detection probabilities due to degradation of environmental samples. Considering this, would it be expected to find potential distinct decay rates and therefore, lowed/increased persistence when comparing the same marker but targeting different regions of distinct fragment sizes?
Figures are relevant to the study and well-designed using adequate statistical approaches. However, for the figures included in the supplemental material, additional descriptions should be provided. Although legends are included in these, it is not clear what they refer to unless searching for information in the main text.
Important to mention that data deposition statement was not found in the main text, and therefore, should be included prior to publication with required information (i.e., raw data accessibility).
Minor comments:
L41: In the abstract, the authors state: “Our findings primarily show that eDNA and eRNA-based detections are consistent for vertebrate and invertebrate taxa.” However, it is not clear what consistency means here. Please clarify.
L387-388: Please add relevant values.

Experimental design

The study is original, with well-defined research questions which were addressed in the manuscript
Methods are described with sufficient detail and information to ensure replicability.

Additional information regarding data deposition and accessibility should be included.

Validity of the findings

The manuscript is clear and uses relevant data collection and statistical analyses to address the main questions. Data is provided and conclusions are clearly stated and properly linked to the study aims.

Reviewer 2 ·

Basic reporting

Clear and unambiguous, professional English are used throughout.
Literature references, sufficient field background/context are provided.
Article structure, figures, tables are appropriate.
Raw data are shared.

Experimental design

I can not evaluate this because of the ambiguous nature of the manuscript.

Validity of the findings

I can not evaluate this because of the ambiguous nature of the manuscript.

Additional comments

I have reviewed the manuscript by Macher et al. The authors performed eDNA and eRNA metabarcoding and compared the results. eRNA metabarcoding has attracted much attention in recent years and its utility and comparison with eDNA is an important research topic. This study is of high importance as it is a study on a topic that fits with current trends. However, the manuscript is poorly presented and it is difficult to understand what experiments were performed. Specific examples are given below.

First, I am very confused about number of samples and number of PCRs performed. Line 113 says that 19 samples were collected at a single site, and line 171 says that a total of 88 PCRs were performed for each type of nucleic acid with duplicates. This seems contradictory. A more careful and precise description would be appreciated.

Second, although I am not a statistician and do not have an alternative, I do not think that the Spearman correlation statistic, as employed in Fig 2 and Fig 5, is appropriate. This is because data from the same sample is used more than once, which I believe does not clear up the issue of multiple comparisons.

Finally, the methods section needs to be carefully checked again. For example, what is TE minimum buffer (L136)? does 10 mM Tris (L141) mean Tris-HCl? If so, what is its pH?

Without these points being corrected, it is impossible for the readers (and reviewers) to understand what has been done and to assess the validity of the results and discussions.

Reviewer 3 ·

Basic reporting

The study compares the effectiveness of eDNA and eRNA metabarcoding in determining the richness and distribution patterns of vertebrate and invertebrate species in a lowland stream in Central Europe. The research shows that more species are detected when using eRNA, and that eRNA detections have a more localized signature compared to eDNA. However, eDNA shows higher species richness for invertebrates. The findings demonstrate that eDNA and eRNA-based detections are consistent for both vertebrate and invertebrate taxa, and that both methods provide important information for biodiversity monitoring and research.

Experimental design

The manuscript should provide a more detailed explanation of the limitations related to the sampling duration and location. For instance, "The sampling process was conducted at a single site and over a short time period, which may limit the generalizability of the obtained data. Therefore, the results should be interpreted carefully within these constraints." Additionally, "Future studies should conduct sampling over different time periods and various sections of the stream to provide a more comprehensive evaluation of species richness and distribution patterns." Such statements should be included to address how these limitations can be mitigated.

Validity of the findings

The statistical analysis methods used should be explained in more detail. For example, "The parameters used in Jaccard distance calculations and how these analyses were performed should be detailed. Additionally, more information should be provided on the selection of independent and dependent variables in the GLM analysis and the appropriateness of the model. This will enhance the reproducibility of the analyses and the reliability of the results.

More information should be provided about potential issues encountered during laboratory procedures (e.g., RNA isolation, DNAse treatment) and how they were addressed. For example, "The challenges faced during the RNA isolation process and the specific protocols implemented to overcome these challenges should be detailed. In particular, the risks of contamination during the DNAse treatment and how these risks were minimized should be explained.

Additional comments

The steps used in eRNA and eDNA analyses should be more clearly stated in the methods section. For example, "All steps used in eRNA and eDNA analyses should be clearly stated along with the reasons for their selection. In particular, detailed explanations of the RNA isolation, DNAse treatment, and reverse transcription processes should be provided. This will enhance the reproducibility and methodological reliability of the study.

---

## Round 0.2 · accepted · Accept

Dear Dr. Macher

I would like to thank you and your co-authors for making the corrections and changes requested by the reviewers. I read and checked your valuable article carefully and am happy to inform you that your article has been accepted for publication in PeerJ.

Reviewer 2 ·

Basic reporting

Authors successfully revised the manuscript. Now I have no concern on the manuscript.

Experimental design

Nothing particular.

Validity of the findings

Nothing particular.

Reviewer 3 ·

Basic reporting

The manuscript is properly organized and the quality of data is reliable. In my opinion, this study is an important contribution to the literature, and it should be accepted in its current form.

Experimental design

The manuscript is properly organized and the quality of data is reliable. In my opinion, this study is an important contribution to the literature, and it should be accepted in its current form.

Validity of the findings

The manuscript is properly organized and the quality of data is reliable. In my opinion, this study is an important contribution to the literature, and it should be accepted in its current form.